# mmWave Zero Order Resonant Antenna with Patch-Like Radiation Fed by a Butler Matrix for Passive Beamforming

**DOI:** 10.3390/s23187973

**Published:** 2023-09-19

**Authors:** Manoj Prabhakar Mohan, Hong Cai, Arokiaswami Alphones, Muhammad Faeyz Karim

**Affiliations:** 1School of Electrical & Electronic Engineering, Nanyang Technological University, Singapore 639798, Singapore; manoj.mohan@ntu.edu.sg (M.P.M.); ealphones@ntu.edu.sg (A.A.); 2Institute for Microelectronic, IME, A*STAR, Singapore 117685, Singapore; 3ETID Department, Texas A&M University, College Station, TX 77843, USA

**Keywords:** CRLH antenna, mmWave antenna, resonant mode, radiation, patch antenna

## Abstract

A small zero-order resonant antenna based on the composite right-left-handed (CRLH) principle is designed and fabricated without metallic vias at 30 GHz to have patch-like radiation. The mirror images of two CRLH structures are connected to design the antenna without via holes. The equivalent circuit, parameter extraction, and dispersion diagram are studied to analyze the characteristics of the CRLH antenna. The antenna was fabricated and experimentally verified. The measured realized gain of the antenna is 5.35 dBi at 30 GHz. The designed antenna is free of spurious resonance over a band width of 10 GHz. A passive beamforming array is designed using the proposed CRLH antenna and the Butler matrix. A substrate integrated waveguide is used to implement the Butler matrix. The CRLH antennas are connected to four outputs of a 4×4 Butler matrix. The scanning angles are $12^{\circ }$, ${-68}^{\circ }$, $64^{\circ }$, and ${-11}^{\circ }$ for excitations from port 1 to port 4 of the 4×4 Butler matrix feeding the CRLH antenna.

## 1. Introduction

Recently, the millimeter wave (mmWave) spectrum has become popular due to its applications in wireless power transfer, synthetic aperture radar imaging, satellite systems, and non-destructive testing [1,2,3]. So, the design of the antennas and beam-forming circuits at mmWave frequencies has also gained much interest.

Techniques to miniaturize antennas have been used for a long time [4]. Most of the techniques fall into two categories: topology-based techniques and material-based miniaturization techniques. Techniques to minimize the size of the wide-band antenna were also reported in [4]. The antenna structures with large dimensions can be miniaturized by bending to form meander lines or broken into small segments using fractals. Even though engineering the ground planes helps reduce the size of the antenna, it also results in an increase in radiation on the back side. Periodic loading, which results in slow wave structures, can also be used to reduce the size of the antenna. Using high-dielectric substrates and superstrates and lowering the dielectric constant using perforation fall under material-based miniaturization. Using magnetodielectric substrates and using piezo electric materials also fall under this category [4]. CRLH concepts have been used in the design of leaky wave antennas and small resonant antennas [5]. The CRLH resonator has resonances in the left-handed region, the right-handed region, and the zero-order resonance. At zero-order resonance, the resonant frequency depends on the parameters of the CRLH resonator and not on the physical dimension of the CRLH resonator. So, a zero-order resonator (ZOR) can be designed at a frequency independent of its physical dimensions [5]. It is an important advantage compared to other miniaturization techniques. The miniaturization using metamaterial also comes under material-based miniaturization of the antennas [4].

The general structure of the CRLH resonator has a metallic via for achieving the left-handed inductance. The shape of the radiation pattern is mainly influenced by the position of the metallic via. So, even though CRLH resonators are smaller in size, the design of the antenna with patch-like radiation is difficult. Several techniques were used in the literature to obtain broad-side radiation using CRLH resonators at zero-order resonance, but monopole like radiation is more common. In [6], monopole-like radiation was achieved using a mushroom-like CRLH resonator and an inductor-loaded resonator. The metallic vias were at the center of the rectangular patch for the CRLH resonator and along the central axis for the inductor-loaded resonator. A patch-like radiation using the CRLH resonator at 4.88 GHz was reported in [7], where metallic-via were replaced by large patches. An eight-cell CRLH resonator arranged in a ring-like fashion with stubs connected at the center was used to design a dual-band antenna [8]. Since the stubs are connected at the center, metallic vias were avoided. Two semicircle patches with shorting pins at one corner formed a two-cell CRLH resonator in [9]. The radiation pattern achieved was patch-like, but if the position of the shorting pin was moved to the center of the semi-circle, monopole-like radiation could be obtained. Broadside radiation was achieved in [10] using a hybrid antenna consisting of a mushroom-type CRLH resonator imbedded inside a rectangular hole of a rectangular patch antenna. The antenna in [10] was fed using a circular patch with a coaxial probe. Via holes were connected to the patch antenna edges through switchable stubs [11] to obtain two types of radiation patterns. If the vias are connected to the patch antenna, negative permittivity is added, and a monopole-like radiation pattern is obtained at zero-order resonance. If the vias were not connected, the antenna in [11] would operate like a patch antenna. A monopolar radiation was obtained for the metamaterial antennas proposed in [12]. To avoid the metallic vias, a few papers designed the resonant antenna based on coplanar waveguide (CPW) [13,14], but CPW configuration increased the back lobes. An artificial magnetic conductor (AMC) screen has been used to minimize the back lobes [14], and it also acts as a separation layer between the human body and the antenna. In the mmWave frequency range, a small change in the dimensions will affect the performance to a greater extent. Therefore, for an antenna at mmWave, it is desirable to design it without metallic vias. A broad-band balun [15] and filter [16] where the CRLH is designed without metallic-via were reported. Only a few works have reported patch-like radiation at zero-order resonance, and that too at the lower frequency range. In this paper, ZOR with patch-like radiation at 30 GHz is designed.

The Butler matrix [17] is one of the methods to realize passive-beam-forming circuits. Since the operating frequency is 30 GHz, the substrate integrated waveguide (SIW) configuration is chosen to design the Butler matrix. The analysis of properties of SIW like dispersion characteristics, propagation constant, modeling, cutoff frequency, and relation between the width of the rectangular waveguide and SIW was reported in [18,19,20]. A review of different microwave circuits implemented on SIW was reported in [21]. The SIW discontinuities were analyzed in [22]. The taper was used in the integration of the microstrip and SIW [23,24] on a single substrate. The different transitions between SIW and other transmission lines were discussed in [25].

There are many beam-forming networks reported, like the Rotman lens [26], Blass matrix [27], Nolan matrix [28], and Butler matrix. The Blass matrix is a lossy network since it requires a direction coupler with a coupling coefficient of one to achieve a lossless Blass matrix [27]. The Nolan matrix is also a type of Blass matrix with losses suppressed, and it also has directional couplers and phase shifters [28]. Among the beam-forming networks, the Butler matrix is chosen because of its simple and modular design. A wide-band Butler matrix in SIW was proposed without crossover in [29]. The comparison between losses in microstrip bends and SIW bends at 77 GHz was also discussed [29]. A compact two-layer SIW Butler matrix was reported in [30] with reduced errors in magnitude and phase. Another two-layer 4×8 Butler matrix in SIW at 38 GHz was proposed in [31] with an antenna array in the third layer, and the crossover was avoided in this reported work also. A 4×8 Butler matrix with a reduced number of crossovers was presented in [32] with amplitude tapering. The number of crossovers was reduced in the two-layer SIW Butler matrix reported in [33] by changing the −3 dB hybrid couplers near the input ports to E-plane hybrid couplers and by reversing the −3 dB hybrid couplers near the output ports. A 7×8 Butler matrix with an antenna array was implanted in three SIW layers in [34] where the boresight beam was created by merging two inputs at the center. The Butler matrix with a modified hybrid coupler was reported in [35]. A mmWave SIW Butler matrix designed at 60 GHz was systematically explained in [36]. The detailed design of the 30 GHz Butler matrix used in this paper is explained in [37]. Metamaterial properties were used to enhance the performance of the antenna implemented in SIW [38]. A power splitter was designed to support an array with two dipole antennas. An epsilon-negative lattice with 3×2 unit cells was used to increase the broadside gain and reduce the cross-section radiation in [38]. Here, the newly designed ZOR antenna is added to the Butler matrix implemented in SIW.

In this paper, a new CRLH resonant antenna at 30 GHz is designed without metallic vias to obtain patch-like radiation. The open-circuited CRLH transmission line is used to design the resonant structure, which is different from the balanced structures reported in [15,16]. The parameter extraction and analysis of CRLH structure using a dispersion diagram are provided along with the simulated and measured results in Section 2. The SIW Butler matrix is used as the feeding structure of the CRLH antenna for beam forming, and the results are discussed in Section 3.

## 2. Design of the CRLH Resonant Antenna

The CRLH resonant antenna is designed as shown in Figure 1. Usually, the CRLH structure is designed using an interdigital capacitor and a short-circuited stub. But the metallic vias in the short-circuited stub prevent the radiation in the broadside direction. To obtain patch-like radiation in the broadside direction, the CRLH resonator is formed without metallic vias. The interdigital capacitor with three fingers followed by a stub with length $l_{1}/2$ forms the first part of the resonator, and this is concatenated with its mirror image to form the CRLH structure. The interdigital capacitor and stub and their *x*-axis mirror image are concatenated to form the CRLH resonator, as shown in Figure 1. This creates a virtual zero at the center of the stub of the resonant antenna. So, the metallic vias needed for the short-circuit stub are removed. The electric field pattern of the CRLH resonant antenna, along with the equivalent circuit, is shown in Figure 2. From the electric field pattern, it can be found that the antenna has an electric zero approximately around the center of the antenna and at that point a virtual ground is formed. The interdigital capacitor with its three fingers is represented by $L_{R_{1}}$ and $C_{L_{1}}$ in the equivalent circuit shown in Figure 2b. The stub is represented by $L_{L_{1}}$ and $C_{R_{1}}$. The mirror image of the interdigital capacitor and short-circuited stubs are also represented as $L_{R_{1}}$ and $C_{L_{1}}$ and $L_{L_{1}}$ and $C_{R_{1}}$, respectively. The electric zero point between the interdigital capacitor and stub and their mirror image is represented as ground in Figure 2b.

The dispersion diagram is obtained by simulating the CRLH unit cell, as shown in Figure 3. The feed lines are de-embedded to obtain the S-parameter response of the CRLH unit cell, and from the S-parameters, the ABCD parameters are found out. The propagation constant is given as cosh⁡γd=(A+D)/2 [39]. The dispersion diagram of the CRLH unit cell is shown in Figure 3. The dispersion diagram shows the CRLH unit cell is unbalanced. The left-handed response is below 30 GHz, and the right-handed response is above 40 GHz. The attenuation gap is between 30 GHz and 40 GHz. The shunt resonance is around 30 GHz, and the series resonance is around 40 GHz. So, the open-circuited resonator formed from this unit cell is resonating at around 30 GHz. The antenna is formed from the open-circuit CRLH resonator.

The antenna is fabricated on Rogers substrate RO4003c with a dielectric constant of 3.55, a loss tangent of 0.0027, and a thickness of 0.508 mm. The photograph of the fabricated structure is shown in Figure 4. The simulated and measured reflection coefficient (S_11_) responses of the antenna are compared in Figure 5. The simulated and measured S parameter responses match well. There is a slight frequency shift, which is due to the fabrication errors. From the measured response, it can be found that the antenna resonates with a central frequency of around 30.77 GHz, and the measured 10 dB fractional bandwidth is 4.2%. The antenna is free of spurious resonant frequencies over the entire band width of 10 GHz. The normalized simulated and measured radiation patterns of the antenna in the YZ ($\varphi =90^{\circ })$ plane are shown in Figure 6. The simulated radiated efficiency of the antenna at the central frequency is approximately 87%. The simulated total gain of the antenna is 5.82 dBi. The measured gain of the antenna is 5.35 dBi. Since the connector is of comparable size to the antenna, during the measurements some of the waves from the transmitting antenna were reflected by the connector, which also contributed to the difference between the simulated and measured radiation patterns in Figure 6 in addition to the fabrication errors.

For the size comparison, the designed antenna is compared with the patch antenna at 30 GHz in Figure 7. The area of the patch antenna is 3.38 mm × 2.545 mm, and the area of the designed CRLH resonant antenna is 2.46 mm × 0.8 mm, which becomes 0.25 $\lambda_{0}$ × 0.08 $\lambda_{0}$ in terms of free space wavelength $\lambda_{0}$. Thus, a reduction of 22.8% is achieved. The comparison of the designed CRLH antenna with the previously reported CRLH resonant antennas is given in Table 1. Patch-like radiation is achieved at the 30 GHz band in this work.

## 3. CRLH Antenna with Butler Matrix

The detailed description of the design of the substrate integrated waveguide (SIW) Butler matrix at 30 GHz is given in [36,37]. Figure 8 shows the block diagram of a 4×4 Butler matrix with four $90^{{}^{\circ}}$ hybrid couplers, two crossovers, two $45^{{}^{\circ}}$ phase shifters, and two $0^{{}^{\circ}}$ phase shifters that are required. Short-slot couplers are used to implement hybrid couplers and crossovers [36].

The short slot coupler is formed by combining two SIWs sideways and removing the central via hole wall to form the slot region. The short slot coupler along with the coupling region are shown in Figure 9. By adjusting the dimensions of the coupling region of the short slot coupler, the ratio of power transmitted to the output ports can be varied. The even and odd-mode analyses are used to analyze the short slot coupler. The ports of the coupler in Figure 9 are denoted as 1 to 4. The ports at the extremes of the coupling regions are denoted by $1^{\prime }$ to $4^{\prime }$ and the reference planes are denoted by dotted lines in Figure 9. Consider an input at port 1. The incident waves for this condition are given as $a=\left[a_{1},\;0,\;0,\;0\right]$. For even- and odd-mode cases, the incident waves are represented as $a_{e}=\left[a_{1}/2,\;0,\;0,\;a_{1}/2\right]$ and $a_{o}=\left[a_{1}/2,\;0,\;0,-a_{1}/2\right]$ respectively and odd and even mode cases are added to obtain the total incident wave, which is $a=a_{e}+a_{o}$. The even- and odd-mode excitations introduce modes ${TE}_{10}$ and ${TE}_{20}$, respectively, in the coupling region. The incident waves from ports $1^{\prime }$ to $4^{\prime }$ are given as ${a^{\prime }}_{e}=\left[{a^{\prime }}_{1}/2,\;0,\;0,\;{a^{\prime }}_{1}/2\right]$ and ${a^{\prime }}_{o}=\left[{a^{\prime }}_{1}/2,\;0,\;0,-{a^{\prime }}_{1}/2\right]$ for the even- and odd-mode excitations. Since ports $1^{\prime }$ to $4^{\prime }$ have a few electrical lengths from ports 1 to 4, the waves ${a^{\prime }}_{e}$ and ${a^{\prime }}_{o}$ are obtained by phase shifting the waves $a_{e}$ and $a_{o}$, respectively. In the odd-mode case, there is an electric wall at the center parallel to the electric walls at the sides in the coupling region. So, the reflected waves from ports $1^{\prime }$ to $4^{\prime }$ are given as ${b^{\prime }}_{o}=\left[0,\;(\frac{{a^{\prime }}_{1}}{2})e^{-j\beta_{20}l},-(\frac{{a^{\prime }}_{1}}{2})e^{-j\beta_{20}l},0\right]$, where $\beta_{20}$ is the phase constant of the ${TE}_{20}$ mode. In contrast to the odd-mode case, for the even-mode excitation, there is a magnetic wall at the center of the coupling region parallel to electric walls at the sides. Because of this, there are small reflections of the incident waves that are cancelled by imposing conditions on the length of the coupling or slot region. The reflected waves from ports $1^{\prime }$ to $4^{\prime }$ are given as ${b^{\prime }}_{e}=\left[0,\;(\frac{{a^{\prime }}_{1}}{2})e^{-j\beta_{10}l},\;(\frac{{a^{\prime }}_{1}}{2})e^{-j\beta_{10}l},0\right]$ with $\beta_{10}$ as the phase constant of the ${TE}_{10}$ mode. The total reflected waves from ports $1^{\prime }$ to $4^{\prime }$ can be found out as $b^{\prime }=\left[0,\;(\frac{{a^{\prime }}_{1}}{2})(e^{-j\beta_{10}l}+e^{-j\beta_{20}l}),\;(\frac{{a^{\prime }}_{1}}{2})(e^{-j\beta_{10}l}-e^{-j\beta_{20}l}),0\right]$. Since the slot or the coupling region considered is symmetric, the S parameters at port 1 are sufficient to fully characterize the region. The S parameters are given as $S_{1^{\prime }1^{\prime }}=0$, $S_{2^{\prime }1^{\prime }}=\cos\left(\left(\beta_{10}-\beta_{20}\right)\frac{l}{2}\right)e^{-j(\beta_{10}-\beta_{20})\frac{l}{2})}$, $S_{3^{\prime }1^{\prime }}=-j\sin\left(\left(\beta_{10}-\beta_{20}\right)\frac{l}{2}\right)e^{-j(\beta_{10}-\beta_{20})\frac{l}{2})}$, and $S_{4^{\prime }1^{\prime }}=0$.

From $S_{3^{\prime }1^{\prime }}$, it can be written as
(1)$$\left(\beta_{10}-\beta_{20}\right)\frac{l}{2}=\sin^{-1}\left|S_{3^{\prime }1^{\prime }}\right|$$

The values of the phase constants $\beta_{10}$ and $\beta_{20}$ are given as
(2)$${\beta_{10}}^{2}=k^{2}-{\left(\frac{\pi }{w}\right)}^{2}$$
and
(3)$${\beta_{20}}^{2}=k^{2}-{\left(\frac{2\pi }{w}\right)}^{2}$$

Substituting (2) and (3) in (1), the equation obtained is
(4)$$\left(\sqrt{\left(k^{2}-{\left(\frac{\pi }{w}\right)}^{2}\right)}-\sqrt{\left(k^{2}-{\left(\frac{2\pi }{w}\right)}^{2}\right)}\right)=\sin^{-1}\left|S_{3^{\prime }1^{\prime }}\right|$$
where k is the wavenumber and w is the width of the slot or the coupling region.

As previously mentioned in the even mode or ${TE}_{10}$, there are reflections and these reflections can be cancelled out if the following approximate condition is satisfied [40]:(5)$$1+e^{-j2\beta_{10}l}=0$$
where l is the length of the slot or the coupling region.

From (5), l can be found out as
(6)$$l=\frac{(2n+1)\pi }{2\beta_{10}}\;$$
where *n* can be zero or a positive integer. Substituting (6) in (4) and substituting the value of $\beta_{10}$, the design equations of the short slot coupler can be found out. A detailed analysis of the short slot coupler can be found in [36,40]. The width and length of the slot region of the short slot coupler are given by
(7)$$w=\frac{\pi }{k}\sqrt{\frac{\left[\pi (2n+1)+4\sin^{-1}\left|S_{3^{\prime }1^{\prime }}\right|\right]\left[3\pi (2n+1)-4\sin^{-1}\left|S_{3^{\prime }1^{\prime }}\right|\right]}{8\sin^{-1}\left|S_{3^{\prime }1^{\prime }}\right|\left[\pi (2n+1)-2\sin^{-1}\left|S_{3^{\prime }1^{\prime }}\right|\right]}}\;$$
(8)$$l=\frac{\left(2n+1\right)\pi }{2\sqrt{k^{2}-{\left(\frac{\pi }{w}\right)}^{2}}}$$
where k is the wavenumber, *n* can be zero or a positive integer, and $S_{3^{\prime }1^{\prime }}$ is the S parameter response in the slot region between ports $3^{\prime }$ and $1^{\prime }$.

For $90^{\circ }$ hybrid couplers, the S parameter response $\left|S_{3^{\prime }1^{\prime }}\right|$ is $1/\sqrt{2}$. Substituting this in (7) and (8) gives the width and length of the slot region to the $90^{\circ }$ hybrid coupler. They are given as [36]
(9)$$w_{1}=\frac{\pi }{k}\sqrt{\frac{4\left(3n+1\right)\left(n+1\right)}{4n+1}}\;$$
(10)$$l_{1}=\frac{\pi }{k}\sqrt{\frac{\left(3n+1\right)\left(n+1\right)}{3}}\;$$

The S parameter responses for the case of crossover are given as $\left|S_{3^{\prime }1^{\prime }}\right|=1$ and $\left|S_{2^{\prime }1^{\prime }}\right|=0$. These values are again substituted in (7) and (8) to find width and length of the slot region of the crossover. They are given as [36]
(11)$$\;w_{2}=\frac{\pi }{k}\sqrt{\frac{\left(6n+1\right)\left(2n+3\right)}{8n}}\;$$
(12)$$l_{2}=\frac{\pi }{k}\sqrt{\frac{\left(6n+1\right)\left(2n+3\right)}{12}}\;$$

For the implementation of $90^{\circ }$ hybrid couplers and crossovers in the Butler matrix, *n* is chosen as 2 in this work. The $90^{\circ }$ hybrid coupler and its dimensions are shown in Figure 10a. The simulated S parameter responses for the port 1 excitation is shown in Figure 10b, along with the phase difference between the S parameter responses $S_{31}$ and $S_{21}$. The simulated magnitude responses of $S_{21}$ and $S_{31}$ are approximately −2.89 dB and −3.53 dB, respectively, at 30 GHz, and the simulated phase difference between them is $S_{31}-S_{21}$ is $90^{{}^{\circ}}$ at 30 GHz. The crossover and its simulated S parameter responses for the input at port 1 are shown in Figure 11. The dimensions of the crossover are also mentioned in Figure 11. As can be found in Figure 11b, only $S_{31}$ is above −3 dB at 30 GHz and all the other S parameter responses are less than −10 dB, so the power is transferred from port 1 to port 3. The simulated insertion loss of $S_{31}$ is approximately 0.53 dB at 30 GHz.

The designed $90^{\circ }$ hybrid coupler and crossover are used to construct the Butler matrix. The phase shifters are designed using the curved SIW lines. The structure of the Butler matrix and a photograph of the fabricated Butler matrix are shown in Figure 12. The simulated and measured S parameter response of the Butler matrix is shown in Figure 13. The simulated phases from ports 5 to 8 for the input at port 1 are approximately $20.82^{\circ }$, ${-9.53}^{\circ }$, ${-38.58}^{\circ }$, and $-92.60^{\circ }$ respectively at 30 GHz. The measured phases from ports 5 to 8 for input at port 1 are approximately ${-26.06}^{\circ }$, ${-15.07}^{\circ }$, ${-44.48}^{\circ }$, and $-170.00^{\circ },$ respectively, at 30 GHz. The final fabricated structure of the Butler matrix with CRLH antenna is shown in Figure 14. The same Rogers RO4003c substrate that was used in the fabrication of the CRLH antenna is also used to fabricate the Butler matrix with the CRLH antenna.

The CRLH antenna described in Section 2 is added as the radiating element for a 4×4 Butler matrix at the four output ports. The CRLH antenna is designed without via holes to obtain the radiation in the broad side direction. The distance between the antennas is 0.75 $\lambda_{g}$. The simulated electric field of the Butler matrix during port 1 excitation is shown in Figure 15. From the electric field distribution, it can be found that the waves from port 1 reach the four output ports at different phases to tilt the radiated beam in the desired direction. The radiation pattern of the antenna in the XZ ($\varphi =0^{{}^{\circ}})$ plane for excitations in the four input ports is shown in Figure 16. The simulated total realized gain is above 7 dBi in all input port excitations, and the simulated scanning angles are $12^{\circ }$ for port 1 excitation, ${-68}^{\circ }$ for port 2 excitation, $64^{\circ }$ for port 3 excitation, and $-11^{\circ }$ for port 4 excitation.

## 4. Conclusions

A CRLH resonant antenna without metallic vias has been designed and experimentally demonstrated to have patch-like radiation in the broadside direction. By utilizing the new CRLH resonator, patch-type radiation can be achieved. The antenna has a low profile and a radiated efficiency of around 87%. The designed antenna has no spurious resonance over a bandwidth of 10 GHz. A passive beamforming array has been designed and fabricated using the Butler matrix and the CRLH antennas. The CRLH antennas are attached to the output ports of the 4×4 SIW Butler matrix, and the performance of the CRLH antenna and the 4×4 SIW Butler matrix is verified. It has potential applications in 5G communication, wireless power transfer, and IoT sensing.

## Figures and Tables

**Figure 1 sensors-23-07973-f001:**
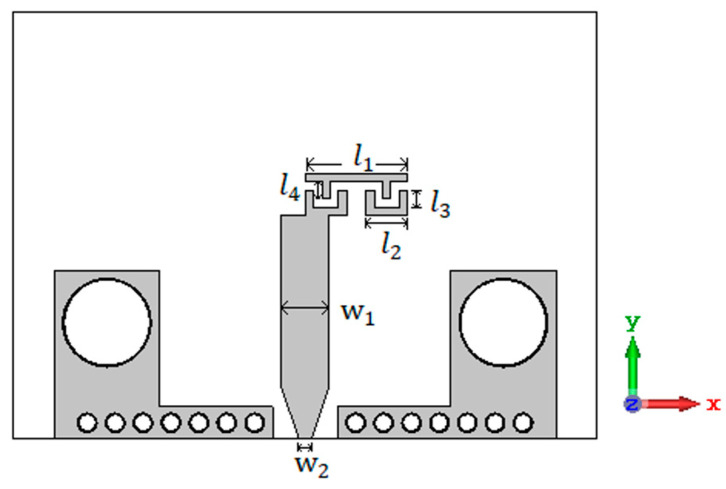
Structure of the CRLH antenna with dimensions $l_{1}$ = 2.46 mm; $l_{2}$ = 1 mm; $l_{3}$ = $l_{4}$ = 0.4 mm; $w_{1}$ = 1.16 mm; $w_{2}$ = 0.3 mm. The width and gap of the interdigital capacitor are 0.2 mm. The two L-shaped structures with two big metallic vias at the sides are for placing the connector.

**Figure 2 sensors-23-07973-f002:**
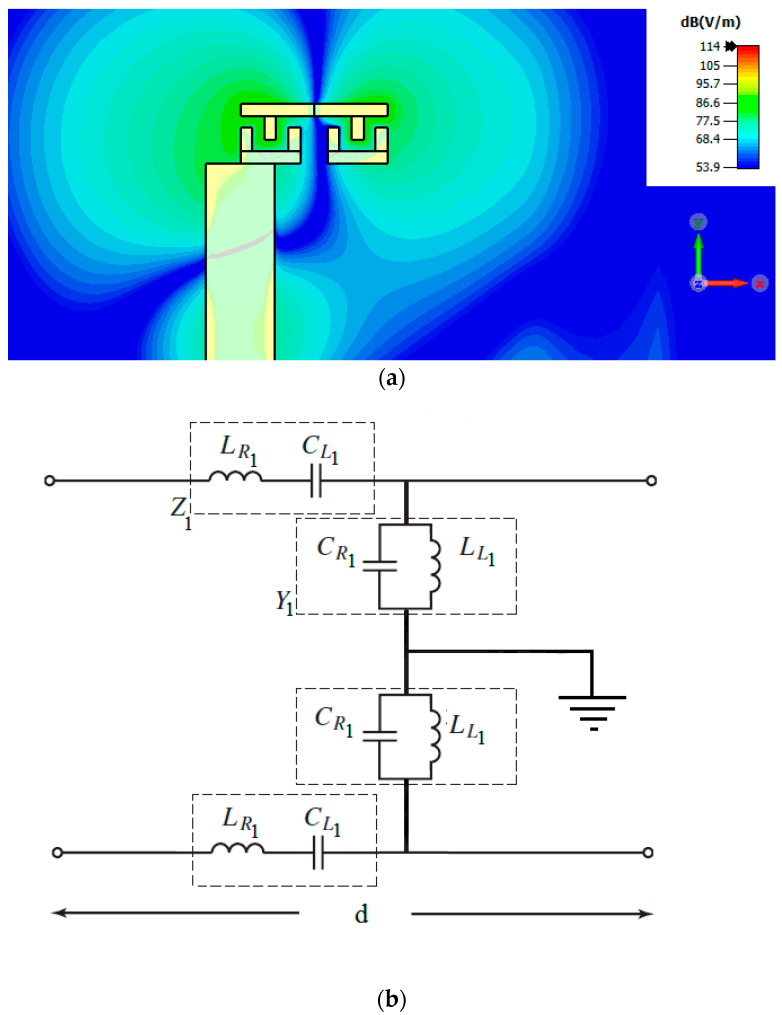
(**a**) Electric field distribution of the antenna; (**b**) equivalent circuit of the CRLH antenna.

**Figure 3 sensors-23-07973-f003:**
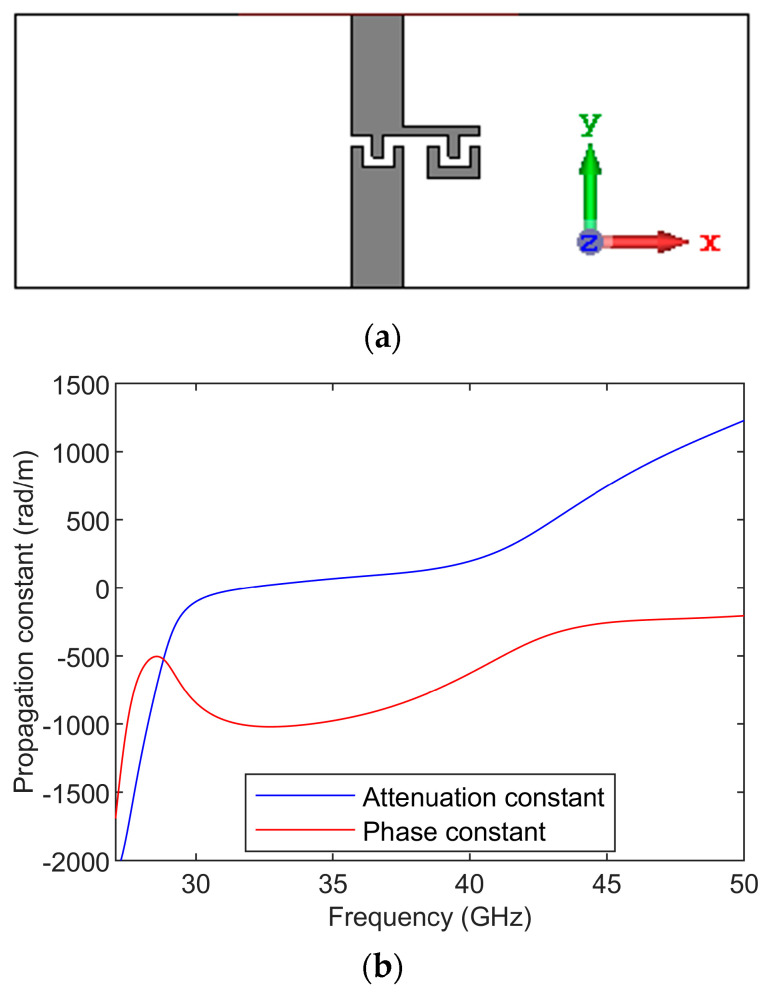
(**a**) Structure of CRLH unit cell used to find the dispersion diagram (**b**) dispersion diagram of the CRLH unit cell used for the antenna.

**Figure 4 sensors-23-07973-f004:**
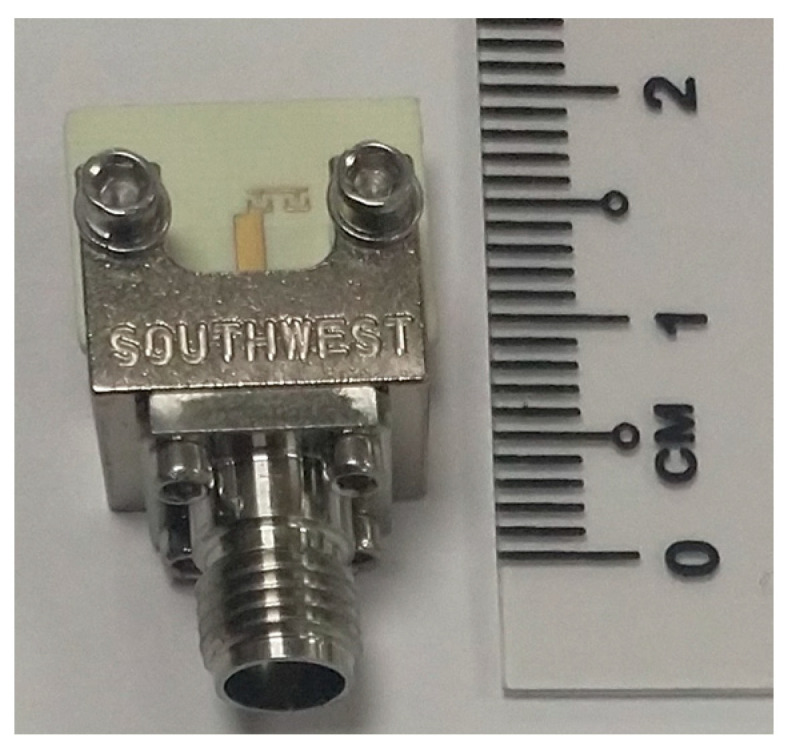
Photograph of the fabricated CRLH antenna.

**Figure 5 sensors-23-07973-f005:**
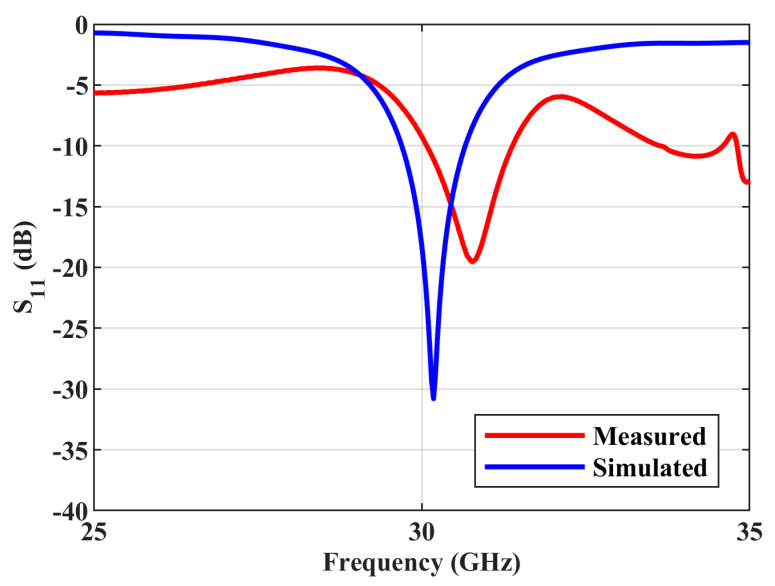
Simulated and measured S parameter response of the antenna.

**Figure 6 sensors-23-07973-f006:**
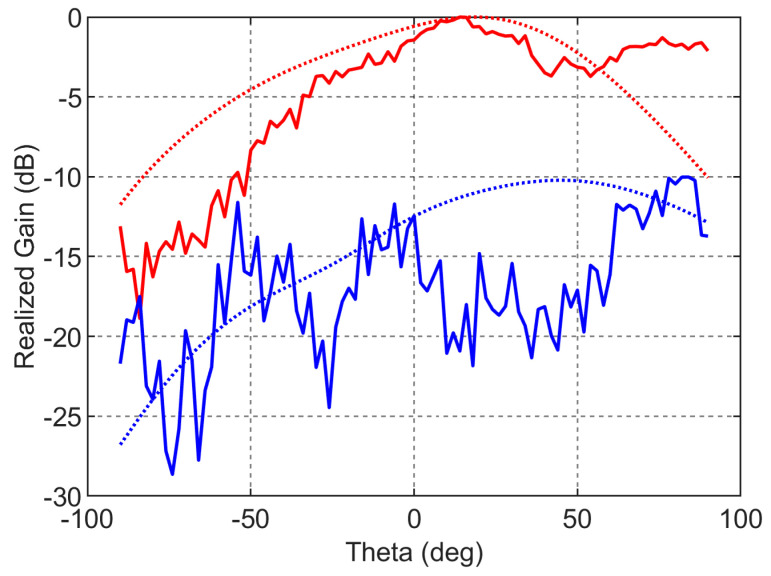
Radiation pattern in the YZ plane. The solid lines denote the measured patterns, and the dotted lines denote the simulated patterns. The red color denotes the co-pol patterns, and the blue color denotes the cross-pol patterns.

**Figure 7 sensors-23-07973-f007:**
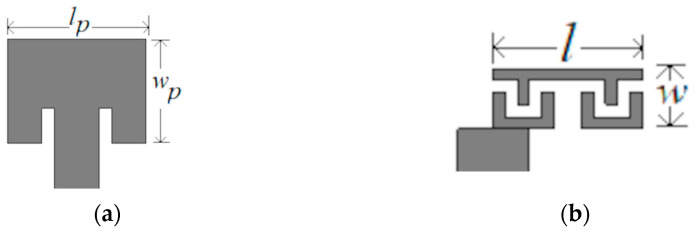
Size comparison. (**a**) Patch antenna at 30 GHz with $l_{p}$ = 3.38 mm and $w_{p}$ = 2.545 mm; (**b**) CRLH resonant antenna with *l* = 2.46 mm and *w* = 0.8 mm.

**Figure 8 sensors-23-07973-f008:**
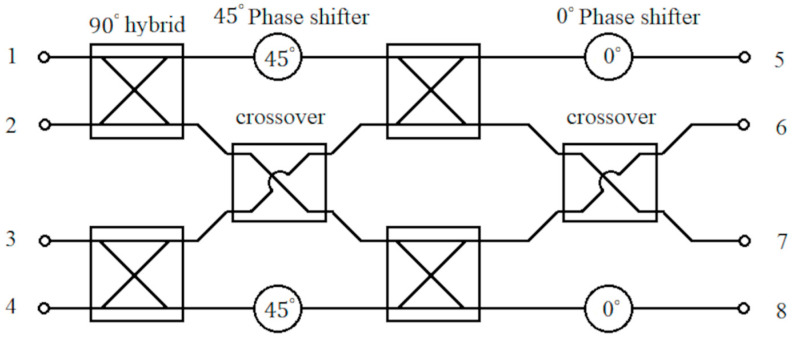
Block diagram of the 4×4 Butler matrix with input ports from 1 to 4 and the output ports from 5 to 8.

**Figure 9 sensors-23-07973-f009:**
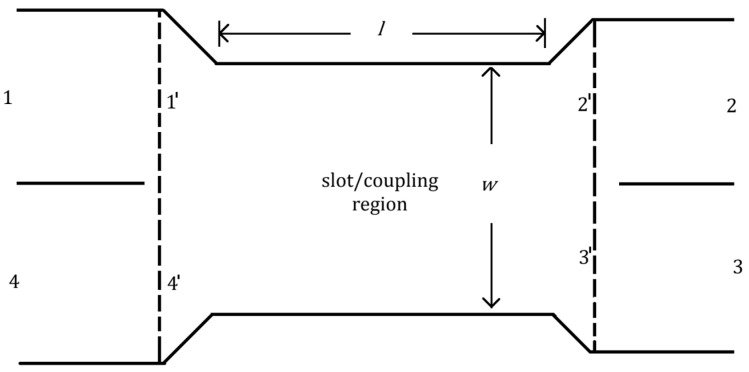
Short clot coupler with the ports 1 to 4 and the ports at the extremes of the coupling region are from $1^{\prime }$ to $4^{\prime }$.

**Figure 10 sensors-23-07973-f010:**
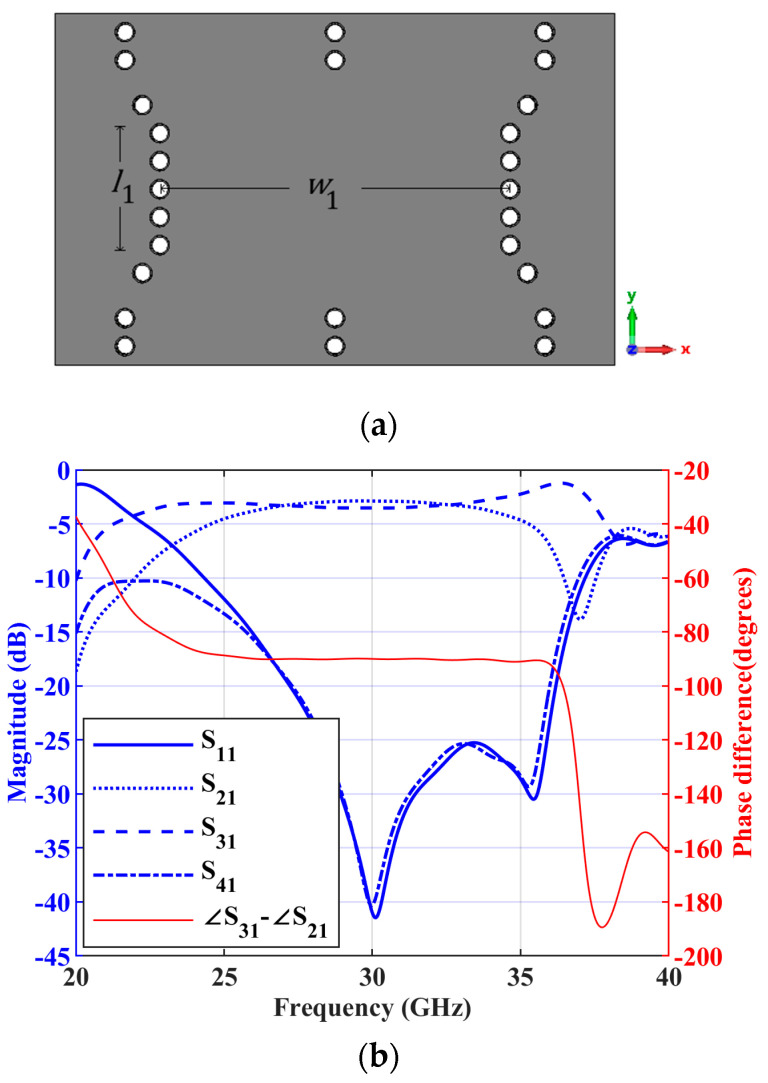
A $90^{\circ }$ hybrid coupler (**a**) structure with dimensions $w_{1}$ = 7.51 mm and $l_{1}$ = 2.4 mm; (**b**) simulated S parameter responses of the hybrid coupler for input at port 1 and a phase difference between $S_{31}\;and\;S_{21}$.

**Figure 11 sensors-23-07973-f011:**
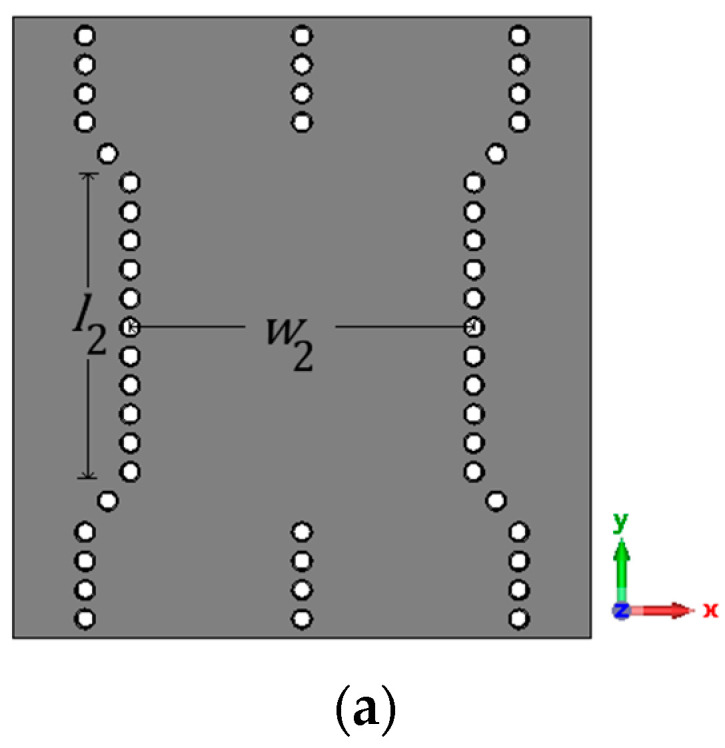
Crossover (**a**) structure with dimensions of $w_{2}$ = 7.13 mm and $l_{2}$ = 6 mm; (**b**) simulated S parameter response of the crossover for the input at port 1.

**Figure 12 sensors-23-07973-f012:**
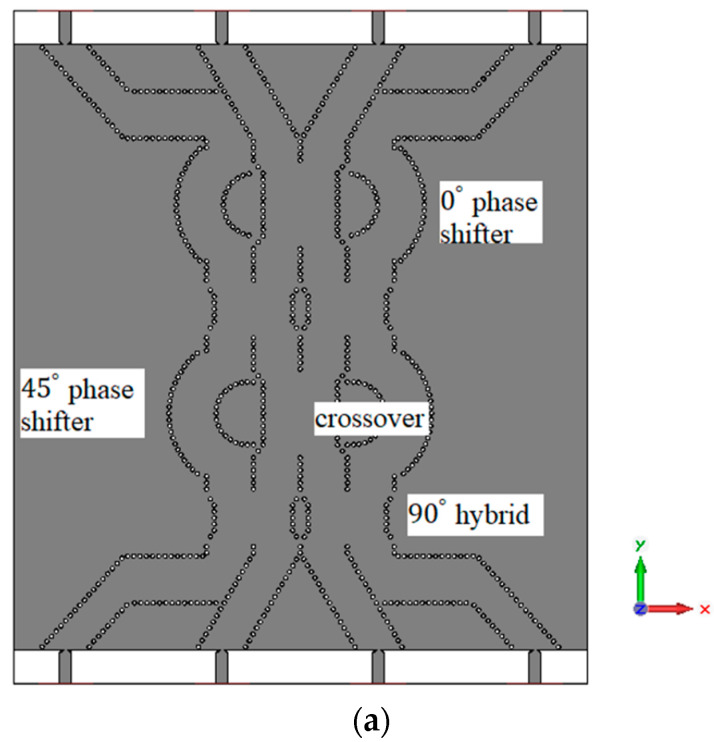
The 4×4 SIW Butler matrix (**a**) structure; (**b**) photograph of the fabricated structure.

**Figure 13 sensors-23-07973-f013:**
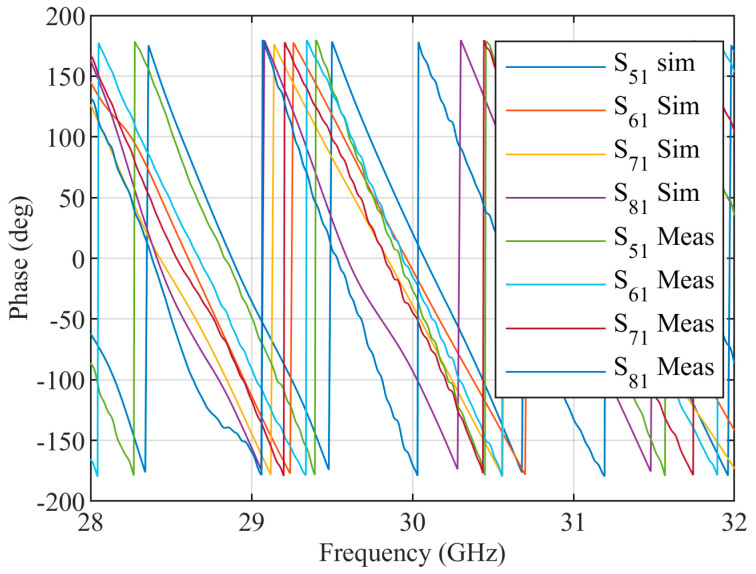
Measured and simulated S parameter responses of the 4×4 SIW Butler matrix for port 1 excitation.

**Figure 14 sensors-23-07973-f014:**
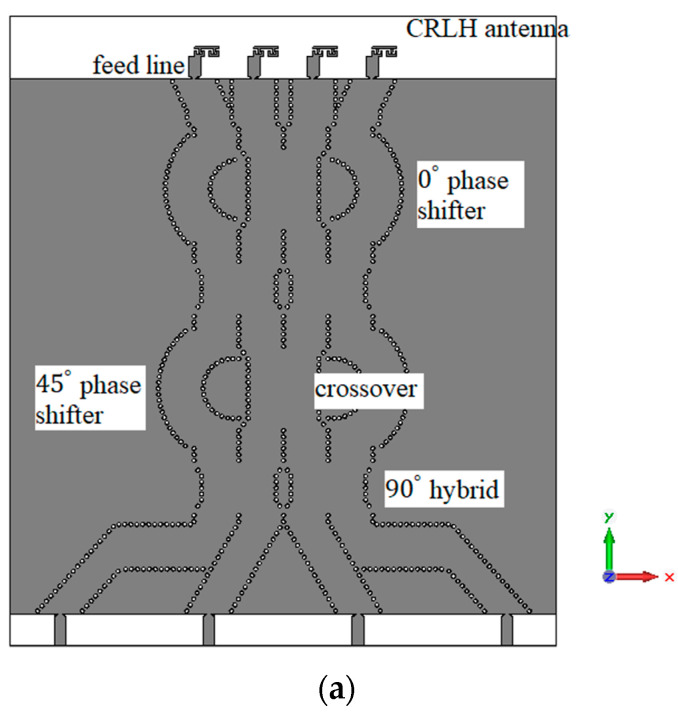
The Butler matrix with a CRLH antenna (**a**) structure; (**b**) photograph of the fabricated structure.

**Figure 15 sensors-23-07973-f015:**
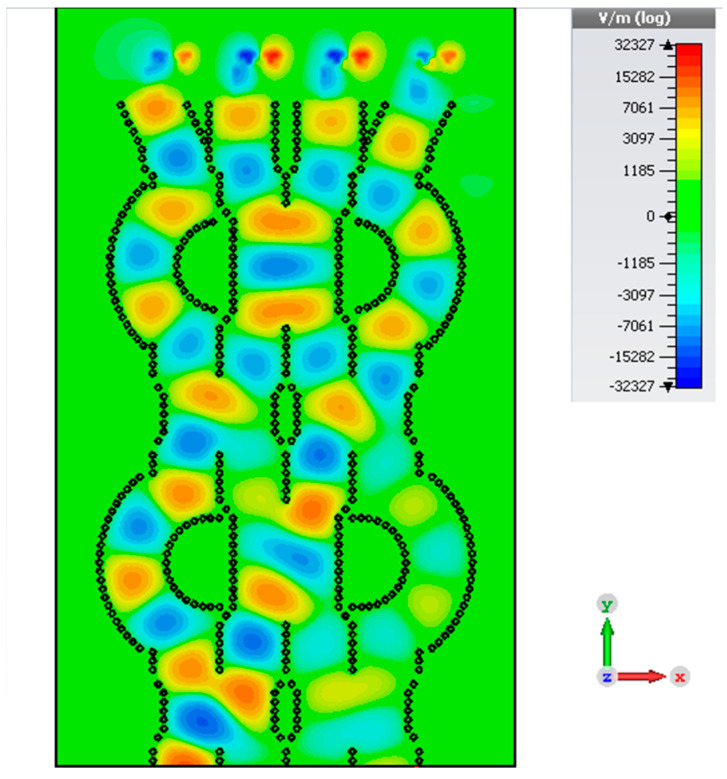
Simulated electric distribution of the Butler matrix with an antenna in SIW.

**Figure 16 sensors-23-07973-f016:**
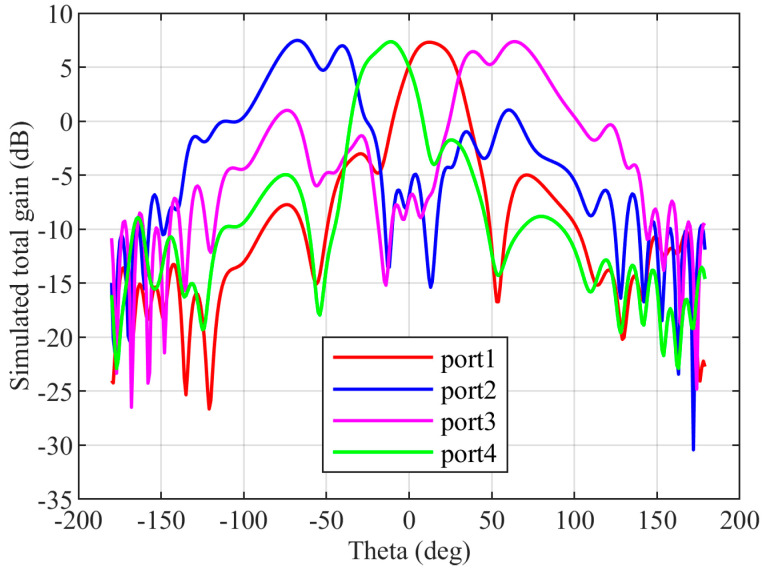
Simulated radiation pattern of the CRLH antenna with the Butler matrix from port 1 to port 4 excitations.

**Table 1 sensors-23-07973-t001:** Comparison of the designed antenna with already published works.

	Frequency of Operation	Antenna Footprint	Radiation Pattern	Gain	Bandwidth %
This work	30 GHz	2.46 mm × 0.8 mm	Patch like	5.35 dBi	4.2%
[6]	3.38 GHz (CRLH) 3.37 GHz (inductor loaded)	$\frac{\lambda_{0}}{6}\times \frac{\lambda_{0}}{6}\times \frac{\lambda_{0}}{53}$	Monopole like	0.87 dBi (CRLH) 0.7 dBi (inductor loaded)	Not mentioned
[7]	4.88 GHz	75% reduction to patch	Patch like	Not mentioned	Not mentioned
[8]	1.93 GHz and 4.16 GHz	Not mentioned	Patch like	−3.21 dBi (lower band), 7.45 dBi (higher band)	Not mentioned
[9]	5.83 GHz	Not mentioned	Patch like	5.06 dBi	Not mentioned
[10]	5.76 GHz	$0.28\;\lambda_{0}\;\times$ $\;0.191\;\lambda_{0}$ $\;\mathrm{and}\;\mathrm{ground}\;\mathrm{plane}\;\mathrm{size}\;0.861\;\lambda_{0}\;\times$ $\;0.766\;\lambda_{0}$	Broadside	4.7 dBi	3.3%
[11]	2.445 GHz	$0.32\;\lambda_{0}\;\times$ $\;0.38\;\lambda_{0}$	Monopole like	3.9 dBi	4.87%

## Data Availability

Not applicable.

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
