# Peer review of "mmWave Zero Order Resonant Antenna with Patch-Like Radiation Fed by a Butler Matrix for Passive Beamforming"

_sensors, 2023, doi:10.3390/s23187973_

Round 1
Reviewer 1 Report
Reviewer Comments:
1. The manuscript's novelty is not clearly presented. It is recommended to emphasize the novelty in both the abstract and introduction sections.
2. Please elaborate on the rationale for choosing a CRLH (Composite Right/Left-Handed) metamaterial structure. Highlight the advantages gained compared to other miniaturized structures.
3. Reframe the introduction to tell a coherent story and explicitly address how your work addresses the gaps in prior research. Specify the problems or limitations your paper aims to overcome.
4. Provide more detailed information about the equivalent circuit and its relationship to the geometry of the structure.
5. Table 1 should include a comparison with recent related works, using up-to-date references. Ensure that size comparisons are made relative to the guided wavelength for fairness.
6. The observed discrepancies between measured and simulated results in Fig. 5 and Fig. 6 appear substantial. Explore factors beyond fabrication tolerance, such as the impact of the end launcher's size relative to the antenna. Consider running additional simulations to investigate this. For further insights, refer to the following source: [Ref] "Performance improvement of substrate integrated cavity fed dipole array antenna using ENZ metamaterial for 5G applications"
7. Equation 1 lacks the left side expression. Please correct this.
8. Create separate figures for each component of the Butler matrix, labeling the lengths (l) and widths (w) obtained from equations 1 to 6. Use different symbols to represent the length and width parameters of each component.
9. Highlight the novel aspects of the proposed Butler matrix in comparison to the one discussed in reference [30].
10. Enhance the readability of all figures by adding a grid.
11. Include measured radiation patterns for all four cases of the Butler matrix.
12. The English language in the manuscript requires significant revision for clarity and coherence.
Please address these comments and make the necessary revisions to the article accordingly.
The English language in the manuscript requires significant revision for clarity and coherence.
Reviewer 2 Report
- In the structure of the CRLH antenna (Fig. 1), the dimensions are small and even significantly less than 1 mm. Have the authors checked the sensitivity of the characteristics on the antenna for manufacturing errors? please comment.
-In fig. 5, the experimental and simulation results do not agree. Although the authors have attributed the shift to manufacturing errors, but it seems more than that. Please clarify the issue. Furthermore Fig. 6 shows severe fluctuations in the experimental results for the gain (specifically for Cross-pol). Has some calibration or environmental factors affected the results?
-The bandwidth in Table 1 has not comparable data from references. This bandwidth can possibly obtained from other data in the references. It would be nice if the authors can extract this comparison for at least one reference if possible.
- Most of the design equations are repeated from Ref. [30]
- There are some typos that can be checked and corrected. For example, in Fig.4, Fig.5, Fig.6,...a space is needed between Fig. and 4, 5, 6,..
Round 2
Reviewer 1 Report
I recommend accepting this manuscript in this form
Author Response
Thank you for your review.
Reviewer 2 Report
-The manuscript is now in a better shape. Please consider the following.
1-In equation (5), "l" is missing in the exponential term. Please make a correction.
2- Table 1 has an overlap with S11 Fig. in the provided file. Please re-check it.
Ok.
